# The citation advantage of linking publications to research data

**Giovanni Colavizza** [1,2], **Iain Hrynaszkiewicz** [3,4], **Isla Staden** [1,5], **Kirstie Whitaker** [1,6], **Barbara McGillivray** [1,6]*

**1** The Alan Turing Institute, London, United Kingdom, **2** University of Amsterdam, Amsterdam, Netherlands, **3** Springer Nature, London, United Kingdom, **4** Public Library of Science, Cambridge, United Kingdom, **5** Queen Mary University, London, United Kingdom, **6** University of Cambridge, Cambridge, United Kingdom

* bmcgillivray@turing.ac.uk

**Data Availability Statement:** Code and data can be found at: https://doi.org/10.5281/zenodo.3470062.

**Funding:** This work was supported by The Alan Turing Institute under the EPSRC grant EP/N510129/1 and by Macmillan Education Ltd, part

## Abstract

Efforts to make research results open and reproducible are increasingly reflected by journal policies encouraging or mandating authors to provide data availability statements. As a consequence of this, there has been a strong uptake of data availability statements in recent literature. Nevertheless, it is still unclear what proportion of these statements actually contain well-formed links to data, for example via a URL or permanent identifier, and if there is an added value in providing such links. We consider 531, 889 journal articles published by PLOS and BMC, develop an automatic system for labelling their data availability statements according to four categories based on their content and the type of data availability they display, and finally analyze the citation advantage of different statement categories via regression. We find that, following mandated publisher policies, data availability statements become very common. In 2018 93.7% of 21,793 PLOS articles and 88.2% of 31,956 BMC articles had data availability statements. Data availability statements containing a link to data in a repository—rather than being available on request or included as supporting information files—are a fraction of the total. In 2017 and 2018, 20.8% of PLOS publications and 12.2% of BMC publications provided DAS containing a link to data in a repository. We also find an association between articles that include statements that link to data in a repository and up to 25.36% (± 1.07%) higher citation impact on average, using a citation prediction model. We discuss the potential implications of these results for authors (researchers) and journal publishers who make the effort of sharing their data in repositories. All our data and code are made available in order to reproduce and extend our results.

## Introduction

More research funding agencies, institutions, journals and publishers are introducing policies that encourage or require the sharing of research data that support publications. Research data policies in general are intended to improve the reproducibility and quality of published research, to increase the benefits to society of conducting research by promoting its reuse, and

of Springer Nature, through grant RG92108 "Effect of data sharing policies on articles' citation counts" granted to BM. Springer Nature provided support in the form of salaries for author IH, but did not have any additional role in the study design, data collection and analysis, decision to publish, or preparation of the manuscript. The specific roles of these authors are articulated in the 'author contributions' section.

**Competing interests:** One of the authors (IH) is at the time of publication in the journal, employed by PLOS, publisher of PLOS ONE. IH was employed by Springer Nature, publisher of the BMC journals, at the time of planning and conducting the research and writing of the original manuscript. This does not alter our adherence to PLOS ONE policies on sharing data and materials. There are no patents, products in development or marketed products associated with this research to declare. All other authors have declared that no other competing interests exist.

**Publisher's Note**: The article involves the independent analysis of data from publications in PLOS ONE. PLOS ONE staff had no knowledge or involvement in the study design, funding, execution or manuscript preparation. The evaluation and editorial decision for this manuscript have been managed by an Academic Editor independent of PLOS ONE staff, per our standard editorial process. The findings and conclusions reported in this article are strictly those of the author(s).

to give researchers more credit for sharing their work [1]. While some journals have required data sharing by researchers (authors) for more than two decades, these requirements have tended to be limited to specific types of research, such as experiments generating protein structural data [2]. It is a more recent development for journals and publishers covering multiple research disciplines to introduce common requirements for sharing research data, and for reporting the availability of data from their research in published articles [3].

Journal research data policies often include requirements for researchers to provide Data Availability Statements (DAS). The policies of some research funding agencies, such as the UK's Engineering and Physical Sciences Research Council (EPSRC), also require that researchers' publications include DAS. A DAS provides a statement about where data supporting the results reported in a published article can be found, whether those data are available publicly in a data repository, available with the published article as supplementary information, available only privately, upon request or not at all. DAS are often in free-text form, which makes it a non-trivial task to automatically identify the degree of data availability reported in them. This is one of the novel contributions of our study. While DAS can appear in different styles and with different titles depending on the publisher, they are a means to establish and assess compliance with data policies [4–6]. DAS are also known as Data Accessibility Statements, Data Sharing Statements and, in this study, 'Availability of supporting data' and 'Availability of data and materials' statements.

Research data policies of funding agencies and journals can influence researchers' willingness to share research data [7, 8], and strong journal data sharing policies have been associated with increased availability of research data [9]. However, surveys of researchers have also shown that researchers feel they should receive more credit for sharing data [10]. Citations (referencing) in scholarly publications provide evidence for claims and citation counts also remain an important measure of the impact and reuse of research and a means for researchers to receive credit for their work.

Several studies explored compliance with journal data sharing policies [11–15]. For example, DAS in PLOS journals have been found to be significantly on the rise, after a mandated policy has been introduced, even if providing data in a repository remains a sharing method used only in a fraction of articles [16]. This is a known problem more generally: DAS contain links to data (and software) repositories only too rarely [17–19]. Nevertheless there are benefits to data sharing [20–22]. It is known that, for example, the biomedical literature in PubMed has shown clear signs of improvement in the transparency and reproducibility of results over recent years, including sharing data [23]. Some previous studies have shown that, mostly in specific research disciplines—such as gene expression studies [24, 25], paleoceanoagraphy [26], astronomy [27] and astrophysics [28]—sharing research data that support scholarly publications, or linking research data to publications, are associated with increased citations to papers [29]. However, to our knowledge, no previous study has sought to determine if providing a DAS, and specifically providing links to supporting data files in a DAS, has an effect on citations across multiple journals, publishers and a wide variety of research disciplines. Making data (and code) available increases the time (and presumably cost) taken to publish papers [30], which has implications for authors, editors and publishers. As more journals and funding agencies require the provision of DAS, further evidence of the benefits of providing them, for example as measured through citations, is needed.

In this study, we consider DAS in journal articles published by two publishers: BMC and PLOS. We focus on the following two questions:

1. are DAS being adopted as per publisher's policies and, if so, can we qualify DAS into categories determined by their contents? In particular, we consider three categories: data

available upon request, data available in the paper or supplementary materials, and data made available via a direct link to it.

2. Are different DAS categories correlated with an article's citation impact? In particular, are DAS which include an explicit link to a repository, either via a URL or permanent identifier, more positively correlated with citation impact than alternatives?

## Materials and methods

### Data

To make this study completely reproducible, we focus only on open access publications and release all the accompanying code (see Data and Code Availability Section). We use the PubMed Open Access (OA) collection, up to all publications from 2018 included [31]. Publications missing a known identifier (DOI, PubMed ID, PMC ID or a publisher-specific ID), a publication date and at least one reference are discarded. The final publication count totals $N = 1,969,175$.

Our analyses focus on a subset of these publications, specifically from two publishers: PLOS (Public Library of Science) and BMC (BioMed Central). PLOS and BMC were selected for this study as they were among the first publishers to introduce DAS. Identifying PLOS journals is straightforward, as the journal names start with 'PLOS', e.g. 'PLOS ONE'. We identify BMC journals using an expert-curated list (see footnote 3 below). We further remove review articles and editorials from this dataset, and are left with a final publication count totalling $M = 531,889$ journal articles. Our data extraction and processing pipeline is illustrated in Fig 1. The processing pipeline, including DAS classification, as well as the descriptive part below were all developed in Python [32], mainly relying on the following libraries or tools: scipy [33], scikit-learn [34], pandas [35], numpy [36], nltk [37], matplotlib [38], seaborn [39], gensim [40], beautifulsoup (https://www.crummy.com/software/BeautifulSoup), TextBlob (https://github.com/sloria/textblob) and pymongo (MongoDB, https://www.mongodb.com).

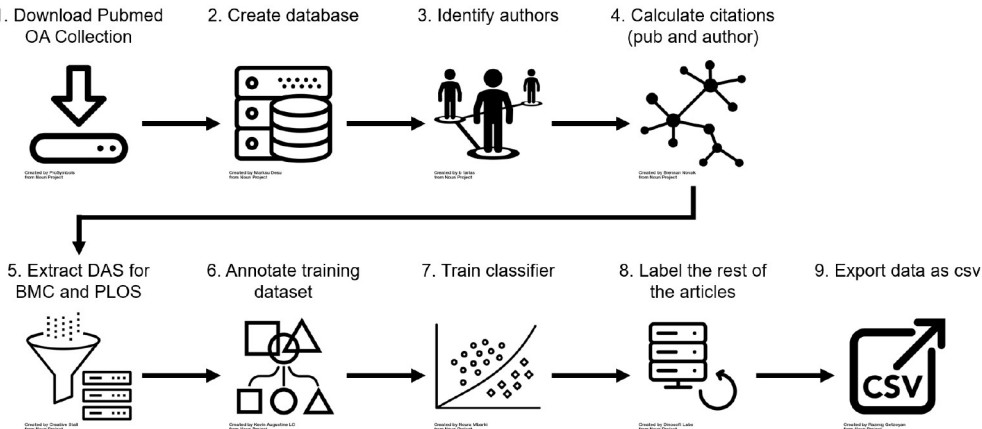

**Fig 1. Data extraction and processing steps.** We first downloaded the PubMed open access collection (1) and created a database with all articles with a known identifier and which contained at least one reference (2; $N = 1,969,175$). Next we identified and disambiguated authors of these papers (3; $S = 4,253,172$) and calculated citations for each author and each publication from within the collection (4). We used these citation counts to calculate a within-collection H-index for each author. Our analysis only focuses on PLOS and BMC publications as these publishers introduced mandated DAS, so we filtered the database for these articles and extracted DAS from each publication (5). We annotated a training dataset by labelling each of these statements into one of four categories (6) and used those labels to train a natural language processing classifier (7). Using this classifier we then categorised the remaining DAS in the database (8). Finally, we exported this categorised dataset of $M = 531,889$ publications to a csv file (9) and archived it (see Data and code availability section below).

## Data availability statements: Policies and extraction

On 1 March 2014, PLOS introduced a mandate which *required* DAS to be included with all publications and required all authors to share the research data supporting their publications [41]. In 2011 BMC journals began to introduce a policy that either *required* or *encouraged* authors to include an equivalent section in their publications, 'Availability of supporting data' [42], and the number of BMC journals that adopted one of these policies increased between 2011 and 2015. In 2015 BMC updated and standardised its policy and all of its journals (more than 250 journals) required—mandated—a DAS (styled as 'Availability of data and materials') in all their publications. This provides sufficient time for publications in these journals to accrue citations for the analysis. Further, all papers published in the BMC and PLOS journals are open access and available under licenses that enable the content and metadata of the articles to be text-mined and analysed for research purposes. We encoded the dates in which these policies were introduced by the different BMC journals, and the type of policy (that is, DAS encouraged or DAS required/mandated) in the list of journals—which also include PLOS journals [43].

The extraction of DAS from the xml files is straightforward for PLOS journals, while it requires closer inspection for BMC journals. We established a set of rules to detect and extract statements from both sets of journals, as documented in our repository. A total of $M_d = 184,075$ (34.6%) publications have a DAS in our dataset. We focus this study on DAS provided in the standard sections of articles according to the publisher styles of PLOS and BMC. While this choice does not consider unstructured statements in publications that might describe the availability of supporting data elsewhere, such as sentences in Methods or Results sections of articles, our analysis intentionally focuses on articles in journals with editorial policies that include the use of a DAS.

## Data availability statements: Classification

The content of DAS can take different forms, which reflect varying levels of data availability, different community and disciplinary cultures of data sharing, specific journal style recommendations, and authors' choices. Some statements contain standard text typically provided by publishers, e.g. 'The authors confirm that all data underlying the findings are fully available without restriction. All relevant data are within the paper and its Supporting Information files.' In other cases, the authors may have decided to modify the standard text to add further details about the location of the data for their study, providing a DOI or a link to a specific repository. Where research data are not publicly available, authors may justify this with additional information or provide information on how readers can request access to the data. In other cases, the authors may declare that the data are not available, or that a DAS is not applicable in their case.

We identified four categories of DAS, further described in Table 1. We use fewer categories than [16], not to impede reliable classification results. Our four categories cover the most well-represented categories from this study, namely: not available or 'access restricted' (our category 0); 'upon request' (our category 1); 'in paper' or 'in paper and SI' or 'in SI' (our category 2); 'repository' (our category 3). We consider category 3 to be the most desirable one, because the data (or code) are shared as part of a publication and the authors provide a direct link to a repository (e.g. via a unique URL, or, preferably, a persistent identifier). We manually categorized 380 statements according to this coding approach, including all statements repeated eight or more times in the dataset (some DAS are very frequent, resulting from default statements left unchanged by authors) and a random selection from the rest. We used a randomly

**Table 1. Categories of DAS identified in our coding approach.**

| Category | Definition | Example |
|---|---|---|
| 0 | Not available | *No additional data available* (common). |
| 1 | Data available on request or similar | *Supporting information is available in the additional files and further supporting data is available from the authors on request* (DOI: 10.1186/1471-2164-14-876). |
| 2 | Data available with the paper and its supplementary files | *The authors confirm that all data underlying the findings are fully available without restriction. All data are included within the manuscript* (DOI: 10.1371/journal.pone.0098191). |
| 3 | Data available in a repository | *The authors confirm that all data underlying the findings are fully available without restriction. The transcriptome data is deposited at NCBI/Gene Bank as the TSA accession SRR1151079 and SRR1151080* (DOI: 10.1371/journal.pone.0106370). |

selected set of 304 (80%) of those statements to train different classifiers and the remaining 76 (20%) statements to test the classifiers' accuracy. The classifiers we trained are listed below:

- NB-BOW: Multinomial Naïve Bayes classifier whose features are the vectors of the unique words in the DAS texts (bag-of-words model);

- NB-TFIDF: Naïve Bayes classifier whose features are the vectors of the unique words in the DAS texts, weighted by their Term Frequency Inverse Document Frequency (TF-IDF) score [44];

- SVM: Support Vector Machines (SVM) classifier [45] whose features are the unique words in the DAS texts, weighted by their TF-IDF score;

- ET-Word2vec: Extra Trees classifier [46] whose features are the word embeddings in the DAS texts calculated using the word2vec algorithm [47];

- ET-Word2vec-TFIDF: Extra Trees classifier whose features are the word2vec word embeddings in the DAS texts weighted by TF-IDF.

TF-IDF is a weighting approach commonly used in information retrieval and has the effect of reducing the weight of words like *the*, *is*, *a*, which tend to occur in most documents. It is obtained by multiplying the term frequency (i.e. the number of times a term *t* appears in a document *d* divided by the total number of terms in *d*) by the inverse document frequency (i.e. the logarithm of the ratio between the total number of documents and the number of documents containing *t*).

We experimented with different parameter values, as detailed below:

- Stop words filter (values: 'yes' or 'no'): whether or not we remove stop words from the texts before running the classifiers. Stop word lists include very common words (also known as function words) like prepositions (*in*, *at*, etc.), determiners (*the*, *a*, etc.), auxiliaries (*do*, *will*, etc.), and so on.

- Stemming (values: 'yes' or 'no'): whether or not we reduce inflected (or sometimes derived) words to their word stem, base or root, for example stemming *fishing*, *fished*, *fisher* results in the stem *fish*.

The best combination of parameter values and classifier type was found to be an SVM with no use of stop words and with stemming, so this was chosen as the model for our subsequent analysis. Its accuracy is 0.99 on the test set, 1.00 only considering the 250 top DAS in the test

**Table 2. Classification report by DAS category.**

| Category | Precision | Recall | F1-score | Specificity | Support |
|---|---|---|---|---|---|
| 0 | 1.00 | 1.00 | 1.00 | 1 | 4 |
| 1 | 1.00 | 1.00 | 1.00 | 1 | 20 |
| 2 | 0.98 | 1.00 | 0.99 | 0.97 | 45 |
| 3 | 1.00 | 0.86 | 0.92 | 1 | 7 |

set by frequency, and the frequency-weighted accuracy is also 1.00. The average precision, recall, and F1-score weighted by support (i.e. the number of instances for each class) are all 0.99. The classification report containing precision, recall, F1-score, and specificity (true negative rate) by category is shown in Table 2. The retained classifier was finally used to classify all DAS in the dataset, keeping manual annotations where available.

## Results

### The presence of data availability statements over time

Fig 2A and 2B show the number of articles in the dataset between the years 2000 and 2018 inclusive. The solid vertical lines show when the publisher introduced a DAS mandate

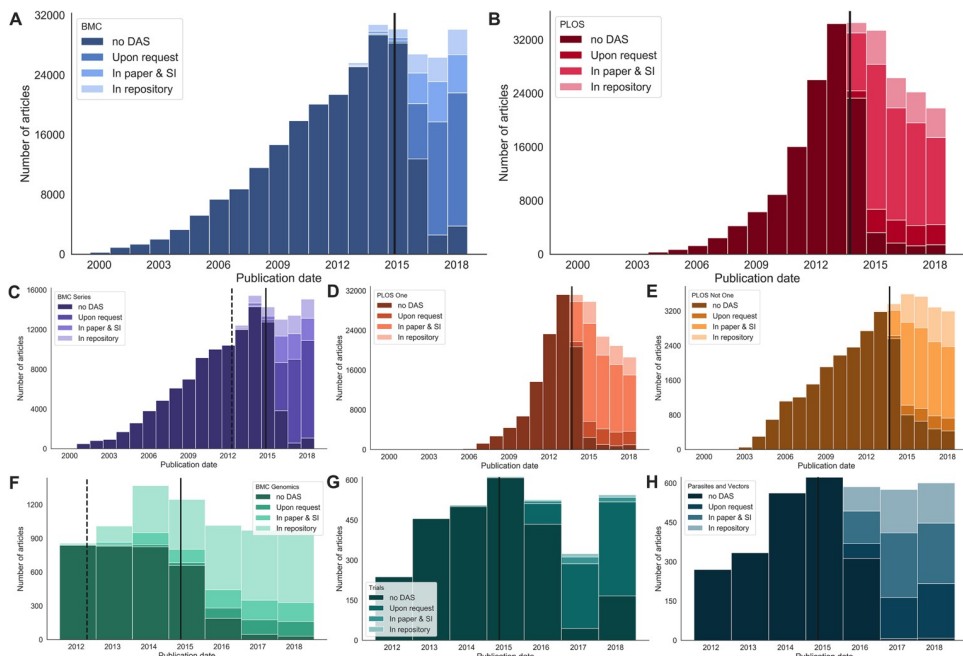

**Fig 2. Data availability statements over time.** All the histograms above show the number of publications from specific subsets of the dataset and classify them into four categories: No DAS (0), Category 1 (data available on request), Category 2 (data contained within the article and supplementary materials), and Category 3 (a link to archived data in a public repository). The vertical solid line shows the date that the publisher introduced a mandated DAS policy. A dashed line indicates the date an encouraged policy was introduced. The groups of articles are as follows. A: all BMC articles, B: all PLOS articles, C: all BMC Series articles, D: *PLOS One* articles, E: PLOS articles not published in *PLOS One*, F: articles from the *BMC Genomics* journal (selected to illustrate a journal that had high uptake of an encouraged policy), G: articles from the *Trials* journal (published by BMC, selected to illustrate a journal that has a very high percentage of data that can only be made available by request to the authors), H: articles from the *Parasites and Vectors* journal (selected to illustrate a journal that has an even distribution of the three DAS categories). Articles are binned by publication year.

(1 May 2015 for BMC, 1 March 2014 for PLOS [41]). BMC journals show a delayed uptake of this policy in published articles, presumably as it was introduced for *submitted* manuscripts rather than accepted manuscripts. The delay accounts for the time these papers would have been undergoing peer review and preparation for publication at the journals. PLOS journals, in comparison and despite PLOS announcing its policy would apply to submitted manuscripts also, appear to have put more effort into early enforcement and therefore have a slightly faster uptake, which may not be accounted for by average submission to publication times. Both publishers show clear adoption of DAS after their introduction of a mandate: in 2018 93.7% of 21,793 PLOS articles and 88.2% of 31,956 BMC articles had data availability statements.

Where the two publishers strongly differ are in the proportion of the different categories of data availability statements. Looking at the two most recent years in the data set, 2017 and 2018, the largest category for BMC (60.0% of 54,719 articles with DAS) is category 1: "*Data available on request or similar*". The remaining BMC articles were 19.2% (10,500 of 54,719) category 2 DAS and 12.2% (6,656 of 54,719) category 3 DAS. Over this same date range, the largest category for PLOS (65.2% of 43,388 articles) is category 2: "*Data available with the paper and its supplementary files*". The remaining PLOS articles were 14.0% (6,065 of 43,388) category 1 DAS and 20.8% (9,013 of 43,388) category 3 DAS. The overrepresentation of categories 1 and 2 for BMC and PLOS articles respectively is likely due to the two publishers having different recommendations in their guidance for authors. For example, 37.3% (40,904 of 109,815) of all PLOS articles which contain a DAS have identical text: "All relevant data are within the paper and its Supporting Information files". We note that although there are an order of magnitude more *PLOS ONE* articles than those published in all other PLOS journals (20,6824 compared to 34,336 in our data set) the pattern of DAS classes are very similar (Fig 2D and 2E). In comparison, the most common DAS (4.8%, 3,594 of 74,260) for BMC is "Not applicable", followed by 3.5% (2,582 of 74,260) which have "The datasets used and/or analysed during the current study are available from the corresponding author on reasonable request." These most common statements are, or have been, included as example statements in guidance to authors, suggesting authors often use these as templates, or copy them verbatim, in their manuscripts. These statements remain the most common in the latest two years in the dataset (2017 and 2018) at 16.5% (PLOS) and 3.6% (BMC) although the proportions have decreased, hopefully indicating a more customised engagement with the data availability requirement by authors. BMC, in 2016, updated its data availability policy including the example DAS statements in its guidance for authors, in conjunction with other journals published by its parent publisher, Springer Nature.

BMC Series journals were *encouraged* to include a section on the "Availability of supporting data" [42] from July 2011. Although the majority of articles published between the encouraged and mandated dates did not have a DAS, 6.0% (1927 of 31,965) of the articles did include this information (Fig 2C). Of these articles with encouraged, rather than mandated, DAS, 65.9% of them (1,270 of 1,927) were category 3: "Data available in a repository". Category 3 DAS are closest to "best practice" data management recommendations, and it is unsurprising that the authors who elected to complete this section when they were not required to do so have shared their data in the most usable manner. Taken together with the most common standard statements described above, we conclude that mandates are beneficial in increasing the number of data availability statements in published articles, but note that they do not guarantee ease of access or re-use.

There are differences in the proportion of DAS classes across academic domains. For example, *BMC Genomics* shows a strong representation of class 3 DAS during both the encouraged and required time periods (Fig 2F). In comparison, and unsurprisingly given the sensitive

nature of the data presented in its articles, the journal *Trials* has a high proportion of class 1 DAS (Fig 2G). We select the journal *Parasites and Vectors* to illustrate that there are also research topics that have high variability of DAS classes within them (Fig 2H).

## Citation prediction

We focus next on predicting citation counts as a means to assess the potential influence of DAS in this respect.

**Dependent variable.** *Citation counts* for each article are calculated using the full PubMed OA dataset (*N* articles above). Citations are based on identifiers, hence only references which include a valid ID are considered. Under these limits, we consider citations given within a certain time-window from each article's publication (2, 3 and 5 years), calculated considering the month of publication, in order to allow for equal comparison over the same citation accrual time (e.g., the three year window for an article published in June 2015 runs to June 2018).

**Independent variables.** We use a set of *article-level variables*, commonly considered in similar studies [48–51]. We include the year of publication, to account for citation inflation over time; the month of publication (missing values are set to a default value of 6, that is June), to account for the known advantage of publications published early in the year; the number of authors and the total number of references (including those without a known ID), both usually correlated with citation impact.

The *reputation of authors* prior to the article publication has also been linked to the citation success of a paper [52]. In order to control for this, we had to identify individual authors, a challenging task in itself [53–57]. We focus on an article-level aggregated indicator of author popularity: the mean and median H-index of an article's authors at the time of publication, calculated from the PubMed OA dataset. In so doing, we minimize the impact of errors arising from disambiguating author names [58, 59], which would have been higher if we had used measures based on individual observations such as the maximum H-index. We therefore used a simple disambiguation technique when compared to current state of the art, and considered two author mentions to refer to the same individual if both full name and surname were found to be identical within all PubMed OA. The total number of authors we individuated with this approach is $S = 4, 253, 172$.

We further consider the following *journal-level variables*: if an article was published by a PLOS or BMC journal; if an article was published under encouraged or required DAS mandates; the domain/field/sub-field of the journal, as given by the Science-Metrix classification [60], in order to control for venue and research area. Despite the fact that journal-level article classifications are not as accurate as citation clustering or other alternatives, the Science-Metrix classification has been recently found to be the best of its class and an overall reasonable choice [61, 62]. We also control for the journal as a dummy variable in some models. An overview of the variables we use is given in Table 3, while a set of descriptive statistics for some of them are reported in Table 4.

The dataset we analyse and discuss here is based on a citation accrualuses a window of three years, and thus includes only publications until 2015 included, in order to allow for all articles to be compared on equal footing. The number of publications under consideration here is thus $M^{2015} = 367, 836$, of which $M_d^{2015} = 45, 968$ with a DAS. Correlation values among a set of variables are given in Table 5, calculated over this specific dataset. While the high correlation between the mean and median H-indexes might indicate multicollinearity, in practice they are not capturing the same signal, as evidenced by similar coefficients when only using one or the other, or both. Results using windows of two or five years, thus considering articles with a corresponding citation accrual time, are consistent.

**Table 3. Summary of variables used in the regression models.**

| Variable | Description | Possible transformations |
|---|---|---|
| $n\_cit_Y$ | Number of citations received within a certain number of years $Y$ after publication. | $ln(n\_cit_Y + 1)$ |
| **Article-level** | | |
| $n\_authors$ | Number of authors. | $ln(n\_authors)$ |
| $n\_references\_tot$ | Total number of references. | $ln(n\_references\_tot + 1)$ |
| $p\_year$ | Publication year. | |
| $p\_month$ | Publication month. | |
| $h\_index\_mean$ | Mean H-index of authors at publication time. | $ln(h\_index\_mean + 1)$ |
| $h\_index\_median$ | Median H-index of authors at publication time. | |
| $das\_category\_simple$ | DAS category (0 to 3. See Table 1). | |
| **Journal-level** | | |
| $is\_plos$ | If PLOS (1) or not (0). | |
| $das\_encouraged$ | If published under an encouraged DAS mandate (1) or not (0). | |
| $das\_required$ | If published under a required DAS mandate (1) or not (0). | |
| $journal\_field$ | Dummy variable, from Science-Metrix. | |

**Model.** The model we consider and discuss here is an Ordinary Least Squares (OLS) model based on the following formula:

$$ln(n\_cit_3 + 1) = ln(n\_authors) + ln(n\_references\_tot + 1) + p\_year + p\_month$$

$$+ ln(h\_index\_mean + 1) + h\_index\_median + das\_category\_simple + is\_plos \qquad (1)$$

$$+ das\_encouraged + das\_required + journal\_field + das\_category * is\_plos$$

An OLS model for citation counts, after a log transform and the addition of 1, has been found to perform well in practice when compared to more involved alternatives [63, 64]. We nevertheless use a variety of alternative models [65], which are made available in the accompanying repository and all corroborate our results. The models we test include ANOVA, tobit and GLM with negative binomial (on the full dataset and on the dataset of papers with 1 or more received citations), zero-inflated negative binomial, lognormal and Pareto 2 family distributions. We further test two mixed effects models nesting articles within journals, one assuming normality on log transformed citation counts (as in the main reported model),

**Table 4. Descriptive statistics for (non-trasformed) model variables over the whole dataset under analysis.**

| Variable/Statistic | Minimum | Median | Mean | Maximum |
|---|---|---|---|---|
| $n\_cit_2$ | 0 | 0 | 0.68 | 166 |
| $n\_cit_3$ | 0 | 0 | 1.13 | 483 |
| $n\_cit_5$ | 0 | 1 | 1.9 | 1732 |
| $n\_cit\_tot$ | 0 | 1 | 2.84 | 2233 |
| $n\_authors$ | 1 | 6 | 6.68 | 2442 |
| $n\_references\_tot$ | 1 | 39 | 41.94 | 1097 |
| $p\_year$ | 1997 | 2014 | 2013 | 2018 |
| $p\_month$ | 1 | 7 | 5.43 | 12 |
| $h\_index\_median$ | 0 | 1 | 1.17 | 28 |
| $h\_index\_mean$ | 0 | 1.2 | 1.56 | 28 |

**Table 5. Correlations among a set of variables.** The values on the top-right half of the table over the diagonal are Spearman's correlation coefficients, the values on the bottom-left half of the table over the diagonal are Pearson's correlation coefficients. All variables are transformed as in the description of the model.

| Variable | $ln(n\_cit_3 + 1)$ | $ln(n\_authors)$ | $p\_year$ | $p\_montd$ | $ln(h\_index\_mean + 1)$ | $h\_index\_median$ | $ln(n\_references\_tot + 1)$ |
|---|---|---|---|---|---|---|---|
| $ln(n\_cit_3 + 1)$ | | 0.16 | 0.14 | -0.02 | 0.25 | 0.2 | 0.22 |
| $ln(n\_authors)$ | 0.16 | | 0.16 | -0.01 | 0.2 | 0.06 | 0.11 |
| $p\_year$ | 0.14 | 0.16 | | -0.02 | 0.39 | 0.32 | 0.1 |
| $p\_month$ | -0.01 | -0.01 | -0.03 | | 0.01 | 0.01 | 0.02 |
| $ln(h\_index\_mean + 1)$ | 0.25 | 0.18 | 0.41 | 0.02 | | 0.85 | 0.12 |
| $h\_index\_median$ | 0.19 | 0 | 0.28 | 0.02 | 0.82 | | 0.08 |
| $ln(n\_references\_tot + 1)$ | 0.24 | 0.15 | 0.13 | 0.02 | 0.14 | 0.07 | |

another assuming lognormality on untransformed citation counts. We also test different model designs, including logistic regression on DAS category and on whether or not a paper is cited at least once. We compare standard OLS and robust OLS here, noting how robust regression results do not differ significantly. Despite robust OLS providing even stronger results for the effect of DAS on citations, we report results from standard OLS in what follows for simplicity. The last interaction term between PLOS and the DAS classification is meant to single out the effect of DAS categories for the two publishers. The results of fitted models are provided in Table 6. All the modelling has been performed in R [66] and RStudio [67], mostly relying on the DMwR [68], glamss [69], mass and nnet [70], vgam [71], ggplot2 [72], tidyverse [73] and stargazer [74] packages.

Regression results point out to a set of outcomes which are known in the literature, namely that articles with more authors and references tend to be slightly more highly cited. We also find a known citation inflation effect for more recent articles (reminding the reader that we consider an equal citation accumulation window of three years overall). Crucially, the mean author H-index is strongly correlated with higher citations, while not so much the median, indicating the preferential citation advantage given to more popular authors. We also find substantial effects at the journal field level, e.g. with General Science and Technology negatively impacting citations (we note that PLOS ONE falls entirely within this category). Articles from PLOS are also, overall, more cited than those from BMC.

Turning our attention to the effect of DAS on citation advantage, we note that the encouraged and required policies play a somewhat minor role. Nevertheless, all DAS categories positively correlate with citation impact, with category 3 standing out and contributing, when present, to an increase of 22.65% (± 0.96%) over the average citation rate of an article after three years from publication, which is 1.26 in the dataset under analysis in this section. The increase is of 25.36% (± 1.07%) considering the 1.13 average citation rate of an article over the whole dataset instead. These positive contributions are less effective for PLOS articles, after controlling for the publisher. When we further control for individual venues (journal), DAS category 3 is the only one remaining significantly correlated with a positive citation impact. Furthermore, we find a minor positive significant interaction (only) between DAS category 3 and the mean author H-index (see repository). These results suggest that the citation advantage of DAS is not as much related to their mere presence, but to their contents. In particular, that DAS containing actual links to data stored in a repository are correlated to higher citation impact. Lastly, we report that the same model (1) just with DAS categories as independent variable shows category 3 with a coefficient of 0.296, which goes to 0.252 in the full model (Table 6), hence corroborating a robust effect. Coefficients are lower and their drop proportionally larger for categories 1 and 2.

**Table 6. OLS and robust LS estimates for the citation prediction model under discussion.** Coefficient standard errors are given in parentheses.

| | Dependent variable: | |
| --- | --- | --- |
| | $ln(n\_cit_3 + 1)$ | |
| | OLS | robust LS |
| | (1) | (2) |
| n_authors | 0.107*** | 0.103*** |
| | (0.002) | (0.002) |
| n_references_tot | 0.197*** | 0.189*** |
| | (0.002) | (0.002) |
| p_year | 0.011*** | 0.011*** |
| | (0.0005) | (0.0005) |
| p_month | −0.011*** | −0.010*** |
| | (0.0005) | (0.0004) |
| h_index_mean | 0.218*** | 0.204*** |
| | (0.004) | (0.004) |
| h_index_median | 0.007*** | 0.008*** |
| | (0.001) | (0.001) |
| C(das_category)1 | 0.085*** | 0.072*** |
| | (0.024) | (0.023) |
| C(das_category)2 | 0.059*** | 0.057*** |
| | (0.019) | (0.018) |
| C(das_category)3 | 0.252*** | 0.271*** |
| | (0.012) | (0.012) |
| C(journal_field)Agriculture, Fisheries & Forestry | −0.066*** | −0.051*** |
| | (0.011) | (0.011) |
| C(journal_field)Biology | -0.009 | 0.007 |
| | (0.009) | (0.009) |
| C(journal_field)Biomedical Research | −0.027*** | −0.012** |
| | (0.005) | (0.005) |
| C(journal_field)Chemistry | −0.242*** | −0.214*** |
| | (0.015) | (0.014) |
| C(journal_field)Clinical Medicine | −0.033*** | −0.021*** |
| | (0.004) | (0.004) |
| C(journal_field)Enabling & Strategic Technologies | 0.047*** | 0.054*** |
| | (0.005) | (0.005) |
| C(journal_field)Engineering | −0.205*** | −0.177*** |
| | (0.019) | (0.019) |
| C(journal_field)General Science & Technology | −0.388*** | −0.370*** |
| | (0.006) | (0.006) |
| C(journal_field)Information & Communication Technologies | 0.007 | 0.025* |
| | (0.013) | (0.013) |
| C(journal_field)Philosophy & Theology | -0.011 | 0.012 |
| | (0.026) | (0.026) |
| C(journal_field)Psychology & Cognitive Sciences | −0.160*** | −0.135*** |
| | (0.021) | (0.021) |
| C(journal_field)Public Health & Health Services | 0.042*** | 0.057*** |
| | (0.006) | (0.006) |
| das_requiredTrue | 0.073*** | 0.070*** |
| | (0.005) | (0.004) |

(*Continued*)

**Table 6.** (Continued)

|  | Dependent variable: | |
|  | $ln(n\_cit_3 + 1)$ | |
|  | OLS | robust LS |
|  | (1) | (2) |
| das_encouragedTrue | −0.052*** | −0.048*** |
|  | (0.004) | (0.004) |
| is_plosTrue | 0.211*** | 0.213*** |
|  | (0.004) | (0.004) |
| C(das_category)1:is_plosTrue | −0.077*** | −0.066*** |
|  | (0.025) | (0.025) |
| C(das_category)2:is_plosTrue | −0.040** | −0.038** |
|  | (0.019) | (0.019) |
| C(das_category)3:is_plosTrue | −0.163*** | −0.192*** |
|  | (0.014) | (0.014) |
| Constant | −22.228*** | −23.297*** |
|  | (0.967) | (0.950) |
| Observations | 367,836 | 367,836 |
| $R^2$ | 0.144 | |
| Adjusted $R^2$ | 0.144 | |
| Residual Std. Error (df = 367808) | 0.593 | 0.665 |
| F Statistic | 2,285.393*** (df = 27; 367808) | |

*p<0.1;
**p<0.05;
***p<0.01

When interpreting these results it should be noted that we consider a relatively small sample of citations compared to the full citation counts of the papers under analysis. We, however, assume that the distribution of citations of the sample is representative of the real citation distribution. We tested this assumption for PLOS ONE publications using Web of Science (WoS). We were able to match 199,304 publications over 203,307 using their DOI. We compared the WoS global citation counts (updated to Summer 2019) with the global citation counts from our dataset (updated to December 2018 included). We find a positive correlation of 0.83 and that citation counts from PubMed OA on average account for 18% of the ones from WoS, supporting our assumption. Based on this assumption, we conclude that there is a positive and significantan up to 25.36% relative gain in citation counts in general for a paper with DAS category 3. We discuss some possible motivations for this effect in our conclusions.

## Discussion

Our study has a set of limitations. First of all, the willingness to operate fully reproducibly has constrained our choices with respect to data. While the PubMed OA collection is sizable, it includes only a fraction of all published literature. Even with respect to indicators based on citation counts (H-index, received citations), we decided not to use larger commercial options such as Web of Science or Scopus. A future analysis might consider much larger citation data, perhaps at the price of full reproducibility. We further focus on DAS given in dedicated sections, potentially missing those given in other parts of an article. Furthermore, we do not assess what a given repository contains in practice: this is *not* a replication study. Finally, citation

counts are but one way to assess an article's impact, among many. These and other limitations constitute potential avenues for future work: we believe that by sharing all our data and code, this study can be updated and built upon for the future analyses.

Future research that evaluates the contents and accuracy of DAS in a more detailed way than in this analysis, e.g., with more sophisticated and granular categorisation of DAS, would be valuable. For example, by comparing whether DAS that are highly templated from journals' guidance for authors are associated with differences in citation counts compared to non-standard statements, and whether DAS are an accurate description of the location of data needed to reproduce the results reported in the article. We assume that non-templated statements imply more consideration of the journal's data sharing policy by the authors, and potentially more rigorous approaches to research data management. However, we found non-templated statements to appear with a lower frequency than statements such as "All relevant data are within the paper and its Supporting Information files".

There are several potential implications of our results. All stakeholders, from funding agencies to publishers and researchers, have further evidence of an important benefit (a potential for increased citations) of providing access to research data. As a consequence, requests for strengthened and consistent research data policies, from research funders, publishers and institutions, can be better supported, enforced and accepted. Introducing stronger research data policies carry associated costs for all stakeholders, which can be better justified with evidence of a citation benefit. Our finding that journal policies that encourage rather than require or mandate DAS have only a small effect on the volume of DAS published will be of interest to publishers, if their goal is to improve the availability of DAS. However, policies often serve to create cultural and behavioural change in a community and to signal the importance of an issue [75], and it is not uncommon for journals and publishers to introduce new editorial policies in a progressive manner, with policies, such as on availability of data and code, increasing in strength and rigour over time. Springer Nature, for example, have indicated they intend to support more of their journals with data sharing policies that do not mandate a DAS to mandate a DAS [3].

Our DAS classification approach, and release of the data and code, may be helpful for stakeholders interested in research data policy compliance, as it enables more automated approaches to the detection, extraction and classification of DAS across multiple journals and publishers, at least in the open access literature. Even wider adoption of DAS as a standard data policy requirement for publishers, funding agencies and institutions would further facilitate the visibility of links to data *as metadata*, enhancing data discoverability, credit allocation and positive research practices such as reproducibility. In fact, machine readable DAS would allow for the development of a research data index extending existing citation indexes and allowing, potentially, to monitor sharing behaviour by researchers and compliance with data policies of different stakeholders. DAS also provides a mechanism for more focused search and enrichment of the literature with links between research data/code, and scholarly articles. Links to research data provided within a DAS are most likely to refer to research data generated by or analysed in a study, potentially increasing the accuracy of services such as EU PubMed Central and Scholarly Link Exchange (Scholix), which can link scholarly publications to their supporting data.

## Conclusion

In this contribution we consider Data Availability Statements (DAS): a section in research articles which is increasingly being encouraged or mandated by publishers and used by authors to state if and how their research data are made available. We use the PubMed Open Access

collection and focus on journal articles published by BMC and PLOS, in order to address the following two questions: 1) are DAS being adopted as per publisher's policies and, if so, can we qualify DAS into categories determined by their contents? 2) Are different DAS categories correlated with an article's citation impact? In particular, are preferred DAS which include an explicit link to a repository, either via a URL or permanent identifier (category 3 in this study) more positively correlated with citation impact than alternatives? These questions are prompted by our intention to assess to what extent open science practices are adopted by publishers and authors, as well as to verify whether there is a benefit for authors who invest resources in order to (properly) make their research data available.

We find that DAS are rapidly adopted after the introduction of a mandate in the journals from both publishers. For reasons in large part related to what is proposed as a standard text for DAS, BMC publications mostly use category 1 (data available on request), while PLOS publications mostly use category 2 (data contained within the article and supplementary materials). Category 3 covers, for both publishers, just a fraction of DAS: 12.2% (BMC) and 20.8% (PLOS) respectively. This is in line with previous literature finding that only about 20% of *PLOS One* articles between March 2014 and May 2016 contain a link to a repository in their DAS [16]. We also note that individual journals show a significant degree of variation with respect to their DAS category distributions.

The results of citation prediction clearly associates a citation advantage, of up to 25.36% (± 1.07%), with articles that have a category 3 DAS—those including a link to a repository via a URL or other permanent identifier, consistent with the results of previous smaller, more focused studies [24–28]. This is encouraging, as it provides a further incentive to authors to make their data available using a repository. There might be a variety of reasons for this effect. More efforts and resources are put into papers sharing data, thus this choice might be made for better quality articles. It is also possible that more successful or visible research groups have also more resources at their disposal for sharing data as category 3. Sharing data likely also gives more credibility to an article's results, as it supports reproducibility [76, 77]. Finally, data sharing encourages re-use, which might further contribute to citation counts.

## Data and code availability

Code and data can be found at: https://doi.org/10.5281/zenodo.3470062 [78].

## Acknowledgments

The authors would like to thank Jo McEntyre and Audrey Hamelers at European Bioinformatics Institute / EUPMC for advice on using their APIs in the planning stage of this study. The authors also thank Angela Dappert at Springer Nature for support in obtaining journal metadata from Springer Nature. GC acknowledges finally thanks James Hetherington, director of Research Engineering at The Alan Turing Institute, for support and advice through the project. GC lastly acknowledges the Centre for Science and Technology Studies, Leiden University, for providing access to their databases.

## Author Contributions

**Conceptualization:** Giovanni Colavizza, Iain Hrynaszkiewicz, Isla Staden, Kirstie Whitaker, Barbara McGillivray.

**Data curation:** Giovanni Colavizza, Iain Hrynaszkiewicz, Kirstie Whitaker.

**Formal analysis:** Giovanni Colavizza, Barbara McGillivray.

**Funding acquisition:** Iain Hrynaszkiewicz, Barbara McGillivray.

**Investigation:** Giovanni Colavizza, Isla Staden, Kirstie Whitaker.

**Methodology:** Giovanni Colavizza, Iain Hrynaszkiewicz, Barbara McGillivray.

**Project administration:** Iain Hrynaszkiewicz, Barbara McGillivray.

**Resources:** Iain Hrynaszkiewicz.

**Software:** Giovanni Colavizza, Isla Staden, Kirstie Whitaker, Barbara McGillivray.

**Supervision:** Barbara McGillivray.

**Visualization:** Isla Staden, Kirstie Whitaker.

**Writing – original draft:** Giovanni Colavizza, Iain Hrynaszkiewicz, Kirstie Whitaker, Barbara McGillivray.

**Writing – review & editing:** Giovanni Colavizza, Iain Hrynaszkiewicz, Isla Staden, Kirstie Whitaker, Barbara McGillivray.

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
