## [Decision Letter · Decision Letter 0]

2 Sep 2019

PONE-D-19-18900

The citation advantage of linking publications to research data

PLOS ONE

Dear Dr Colavizza,

Thank you for submitting your manuscript to PLOS ONE. After careful consideration, we feel that it has merit but does not fully meet PLOS ONE’s publication criteria as it currently stands. Therefore, we invite you to submit a revised version of the manuscript that addresses the points raised during the review process.

I have asked two experts to comment on your submission and also read your submission with great interest. Both the reviewers and I agree that your work is timely and important and that it sheds important light on journal data policies and their relation with data sharing and eventual impact as reflected in citations. However, you need to consider carefully the major statistical issues raised by the second reviewer, and several minor issues concerning details of the data, methods, and results.

MINOR ISSUES

The first reviewer asks for clarifications and details to be added to different sections of your manuscript. This reviewer also rightly notes that the use of causal terms is not warranted given the correlational data that you used. The second reviewer also provided useful feedback (als in his attached PDF) that could help you improve the writing and details of the manuscript.  Finally, I wondered myself whether you could add more details on the citation data in relation to the well known Scopus and ISI Web of Science databases.

MAJOR ISSUE

The second reviewer was rather critical of your analyses. I agree with this reviewer that many different models could potentially be run and that the choice of analysis can affect the key result of the citation advantage. It is important that you do not oversell the results (not necessary here at PLOS) and that you avoid the issue of overfitting (or any selection of models that could inflate the citation advantage). Moreover, I agree with this reviewer that given the typically very skewed distribution of citation data, it would be better to use negative binomial regressions (or a zero-inflated Poisson model) instead of normal regressed based on the log of number of citations (plus 1). I ask you to consider carefully the statistical issues raised by this reviewer and re-run the key models to ensure that the outcomes are robust agains the use of alternative statistical and predictive models.

We would appreciate receiving your revised manuscript by Oct 14 2019 11:59PM, but please let us know if you need more time. To enhance the reproducibility of your results, we recommend that if applicable you deposit your laboratory protocols in protocols.io, where a protocol can be assigned its own identifier (DOI) such that it can be cited independently in the future. For instructions see: http://journals.plos.org/plosone/s/submission-guidelines#loc-laboratory-protocols

We look forward to receiving your revised manuscript.

Kind regards,

Jelte M. Wicherts

Academic Editor

PLOS ONE

Journal Requirements:

2. We note that Iain Hrynaszkiewicz is affiliated with the Public Library of Science (PLOS). To ensure transparency, please include this information in the Competing Interests statement.

This work was supported by The Alan Turing Institute under the EPSRC grant EP/N510129/1 and by Macmillan Education Ltd, part of Springer Nature, through grant RG92108 "Effect of data sharing policies on articles' citation counts" both granted to BMC and IH.

We note that one or more of the authors are employed by a commercial company: Macmillan Education Ltd

Reviewers' comments:

Reviewer's Responses to Questions

**Comments to the Author**

1. Is the manuscript technically sound, and do the data support the conclusions?

Reviewer #1: Yes

Reviewer #2: Partly

2. Has the statistical analysis been performed appropriately and rigorously? 

Reviewer #1: Yes

Reviewer #2: No

3. Have the authors made all data underlying the findings in their manuscript fully available?

Reviewer #1: Yes

Reviewer #2: Yes

4. Is the manuscript presented in an intelligible fashion and written in standard English?

Reviewer #1: Yes

Reviewer #2: Yes

5. Review Comments to the Author

Reviewer #1: This article describes a rigorous and technically sound research effort to examine the content of a large sample of data availability statements (DAS) reported in PLOS and BMC journals and their association with citation impact. It is highly commendable that the article is accompanied by a detailed repository containing the research data and analysis scripts which should enable verification and re-use (note that other than a cursory examination, I have not actually tried to use these files myself). Limitations of the research design are discussed forthrightly in the discussion section and generally the conclusions seem well calibrated to the research design and the evidence obtained. I only have minor suggestions to improve the clarity of reporting.

“DAS in PLOS” – should the specific PLOS journal be specified?

I suggest that the abstract and introduction mention briefly describe the use of a classifier was used as this was a key part of the method.

“We established a set of rules to detect and extract statements from both sets of journals, as documented in our repository” – it is not obvious where to find the relevant details in the repository and more details in manuscript would be helpful. The repository is commendably detailed, but linking directly to the relevant section will help readers find the correct information more quickly.

“We identified four categories of DAS, further described in Table 1. We use fewer categories than [16], mainly due to the sparsity of most of them” – the referenced study is based on PLOS, is there good reason to believe these statement types are also sparse in the BMC dataset?

Description of classifier results – Table 2 and corresponding text – I think further explanation would be very helpful here. I don’t think the headings in Table 2 are clearly defined and the text refers to ‘accuracy’ which doesn’t appear in the table. As far as I can tell, the results pertain to sensitivity, but not specificity i.e., do we know how many DAS the classifier might miss? Also, if I have understood correctly, the training set and test set were specifically populated with DAS from four pre-selected categories. If so, how do we know how the classifier will perform in the wild? (i.e., when handling other DAS types).

Figure 2 – I suggest the legend includes actual category descriptions/names rather than “class 1” etc. Otherwise the reader has to keep moving between the main text and the figure.

Generally, the conclusions are well calibrated to the evidence. However, occasionally, language that implies a causal relationship is used when that is not warranted by the present research design e.g., “all DAS categories positively impact citation counts”

Reviewer #2: Review of PONE-D-19-18900

This is a review of Colavizza et al “The citation advantage of linking publications to research data”, as submitted to PLOS ONE. The article addresses the hypothesis that the type of Data Availability Statement (DAS) included at the end of a published paper affects its subsequent citation rate.

The authors gather a very large dataset of Open Access articles from the PubMed OA collection, all from either PLOS or BMC journals. They use Natural Language Processing to extract the DAS and then classify them into four categories according to their level of detail. The authors then apply a linear model that includes many other variables known to influence citation count (e.g. time since publication, author popularity, and manuscript field). After accounting for these other variables they find a small but significant boost from including a DAS, particularly the most detailed category (3).

This is a decent article and will become a useful contribution, but I would like the authors to address the following points.

First, the model fitting is the weakest aspect of the article. The red flag for me is in the abstract, “can have up to 25.36% higher citation impact”, as this phrasing sounds like the authors tried many different models before arriving at one that shows the largest effect of DAS on citations. This might also explain why the model presented in the article contains both h_index_mean and h_index_median, but leaves out journal. I appreciate that decisions about model building are difficult, and reviewers always want some variable included/excluded, but the authors need to avoid the appearance of p-hacking; this is best done by deciding exactly what question(s) the article is testing and focus on one or two models that test it most efficiently (even if the outcome is less compelling).

A more robust approach that is used in the medical literature for similar circumstances (i.e. observational data) is as follows. First, estimate the strength of the relationship between your central variable of interest (DAS category) and citations. This will approximately estimate the amount of variance in citations that DAS category could explain. Next, present the full model, including all the covariates that could also affect citation count, either independently (e.g. year) or through their effects on DAS (e.g. H-index, where more meticulous researchers have both better DAS and better articles, and hence more citations). The effect size associated with DAS category will certainly go down, but what matters here is by how much: a large drop implies that the other covariates were driving most of the initial effect size, whereas a small decrease shows that DAS category itself has a robust effect on citations.

The authors should be also using a mixed effects model, where articles are nested within journal, allowing journal identity to act as a random effect. This approach also eliminates the need for a separate subject area and publisher variables. The extremely wide scope of PLOS ONE presents some difficulties, but the authors could consider splitting PLOS ONE articles up by their broad subject category (e.g. PLOS_ONE_biology_and_life_sciences), and using this as journal identity for these articles instead.

Since citations are count data and the mean is close to zero (Table 4), the model should be using a Poisson distribution and not a normal distribution. Using a normal model under these circumstances may lead to odd results, such as predicting negative citations (which are clearly impossible). If the majority of articles have zero citations, it may be better to use a negative binomial model or even a zero-inflated Poisson model. Moreover, x-axis variables should remain untransformed unless there is a specific a priori reason to transform them, or if transforming them corrects a serious issue with the distribution of the residuals: one cannot put in both the transformed and the untransformed covariates and pick the one that looks best by p-value. Note that it’s the model residuals that need to be normally distributed, and not the covariates themselves.

The authors may also want to consider including subsequent analyses that home in on particular subsets of the data. For example, the types of DAS encountered seems to differ quite significantly between the ‘encouraged’ and the ‘mandatory’ time period, with high quality category 3’s dominating the former. These category 3 DAS likely come from competent, highly motivated groups, and these are expected to get more proportionally more citations even in the absence of the DAS. Analysing the ‘encouraged’ and ‘mandatory’ data separately would help readers better understand this difference. As it is, I was left with the suspicion that the significant citation advantage for the category 3 DAS in the main model is driven principally by the ‘encourage’ part of the dataset.

I was also (initially) puzzled why the authors include year and month of publication as separate variables. They state that there is a “known advantage of publications published early in the year” (line 243), but don’t provide a reference. Is this advantage above and beyond that expected because an article published in January is 11 months older than one published in December of the same year? If this advantage really does exist, does the reader really need to worry about it? My initial idea was to recommend having ‘months since publication’ as a single variable, as it’s simpler and easier to understand, but it seems that the citation data are calculated only for articles that appear 36 months after publication of the focal article, so ‘months since publication’ is constant. It would be worth making this a bit clearer in the article. Minor point: does the number of citations in the dataset grow each year just because the number of OA articles published each year is growing? Or is the number of references in each article increasing too?

I’ve made a number of other suggestions directly onto the pdf.

6. PLOS authors have the option to publish the peer review history of their article (what does this mean?). If published, this will include your full peer review and any attached files.

Reviewer #1: No

Reviewer #2: No

---

## [Decision Letter · Decision Letter 1]

19 Dec 2019

PONE-D-19-18900R1

The citation advantage of linking publications to research data

PLOS ONE

Dear Dr Colavizza,

Thank you for submitting your revised manuscript to PLOS ONE. Please accept my apologies for the delay in handling your manuscript which was caused by a host of events including sickness in my family and the fact that I was in the midst of moving houses.

After careful consideration, we feel that your revised submission can form an interesting addition to the literature on the citation advantage of data sharing, but that it does not yet fully meet PLOS ONE’s publication criteria as it currently stands. Therefore, we invite you to submit a revised version of the manuscript that addresses the points raised during the review process.

Although the reviewers and I agree that your revisions have strengthened the manuscript, the second reviewer continues to be critical of several key issues in the presentation of results and the analyses you reported. I agree with this critique, and so It is crucial that you do not oversell your results and that you consider carefully the statistical points raised by this reviewer. It is particularly important to address the last point on the real possibility that your results are not entirely robust against alterations in the statistical model. Given that you did not pre-register your analyses, the best approach here is to add additional sensitivity analyses to an (online) appendix and discuss these results in an open manner in your discussion, even if they turn out to show several results not to be entirely robust against alterations in your statistical model. It is better to be safe (and open) than sorry on this issue.

We would appreciate receiving your revised manuscript by Jan 30 2020 11:59PM. To enhance the reproducibility of your results, we recommend that if applicable you deposit your laboratory protocols in protocols.io, where a protocol can be assigned its own identifier (DOI) such that it can be cited independently in the future. For instructions see: http://journals.plos.org/plosone/s/submission-guidelines#loc-laboratory-protocols

We look forward to receiving your revised manuscript.

Kind regards,

Jelte M. Wicherts

Academic Editor

PLOS ONE

Journal Requirements:

Note from Editorial Staff: Note that if the manuscript is accepted for publication, the following Publisher's Note will be added:

'Publisher's Note: The article involves the independent analysis of data from publications in PLOS ONE. PLOS ONE staff had no knowledge or involvement in the study design, funding, execution or manuscript preparation, although one of the authors has since been employed by PLOS whilst the manuscript was under consideration for publication. The evaluation and editorial decision for this manuscript have been managed by an Academic Editor independent of PLOS ONE staff, per our standard editorial process. The findings and conclusions reported in this article are strictly those of the author(s).'

If you have any questions about this, please feel free to contact our office at plosone@plos.org.

Reviewers' comments:

Reviewer's Responses to Questions

**Comments to the Author**

1. If the authors have adequately addressed your comments raised in a previous round of review and you feel that this manuscript is now acceptable for publication, you may indicate that here to bypass the “Comments to the Author” section, enter your conflict of interest statement in the “Confidential to Editor” section, and submit your "Accept" recommendation.

Reviewer #1: All comments have been addressed

Reviewer #2: (No Response)

2. Is the manuscript technically sound, and do the data support the conclusions?

Reviewer #1: Yes

Reviewer #2: Partly

3. Has the statistical analysis been performed appropriately and rigorously? 

Reviewer #1: Yes

Reviewer #2: No

4. Have the authors made all data underlying the findings in their manuscript fully available?

Reviewer #1: Yes

Reviewer #2: Yes

5. Is the manuscript presented in an intelligible fashion and written in standard English?

Reviewer #1: Yes

Reviewer #2: Yes

6. Review Comments to the Author

Reviewer #1: Most of my comments have been adequately addressed. I have no further comments.

Reviewer #2: Review of PONE-D-19-18900R1 “The citation advantage of linking publications to research data”

This article has been revised in response to the first round of peer review. I’m largely supportive of publication at this point, but I still (strongly) object to the authors’ use of the phrase ‘can have up to’ in the abstract – by focusing their most sensational result they are not accurately reporting what they found.

They should follow their own suggestion in the response to reviewers: “We believe that quantifying the average increase in citations a paper with a DAS has in the abstract is quite important to convey the whole point of the analysis” and focus on the average increase in citations that they found, not the largest increase. Expressing it as a percentage also obscures the relatively small effect size (approx. 1.51 citations after three years with a type 3 compared to 1.14 with type 0)

There are still lingering issues with the statistics. At this point I’m fairly sure they don’t undermine the main result, but it would good to have them addressed.

First, the mean and median of the h-index are closely correlated (0.82 in Table 5) so the authors are not convincing when they claim they are capturing a different signal (lines 264 to 267). Given the known difficulties in making sense of coefficients for closely correlated variables, they should also refrain from over-interpreting this aspect of the results (“Crucially, the mean author H-index is strongly correlated with higher citations, while not so much the median, indicating the preferential citation advantage given to more popular authors.” Lines 283-284).

Second, including an interaction term in the model (das_category*is_plos) means that the coefficients for these variables alone (das_category and is_plos) no longer have a simple interpretation. For example, the coefficient 0.252 for C(das_category)3 in Table 6 only applies to when is_plos is 0 (i.e. just BMC articles). Since the das_category is their main focus, the authors need to address this throughout.

Third, I still disagree with the authors’ philosophical approach to their analyses and would much prefer that they decide on their statistical model before they collect their data and stick with that. Analysing their data in lots of different ways with many different variables just looks like p-hacking, even if that’s not how the authors perceive it.

7. PLOS authors have the option to publish the peer review history of their article (what does this mean?). If published, this will include your full peer review and any attached files.

Reviewer #1: No

Reviewer #2: No

---

## [Editor Report · Decision Letter 2]

2 Mar 2020

The citation advantage of linking publications to research data

PONE-D-19-18900R2

Dear Dr. Colavizza,

We are pleased to inform you that your manuscript has been judged scientifically suitable for publication and will be formally accepted for publication once it complies with all outstanding technical requirements.

With kind regards,

Jelte M. Wicherts

Academic Editor

PLOS ONE
---

## [Editor Report · Acceptance letter]

1 Apr 2020

PONE-D-19-18900R2 

The citation advantage of linking publications to research data 

Dear Dr. McGillivray:

I am pleased to inform you that your manuscript has been deemed suitable for publication in PLOS ONE. Congratulations! Your manuscript is now with our production department. 

With kind regards,

on behalf of

Dr. Jelte M. Wicherts 

Academic Editor

PLOS ONE